# Model evidence for a seasonal bias in Antarctic ice cores

Michael P. Erb[1,2,3], Charles S. Jackson [1], Anthony J. Broccoli[4], David W. Lea[5], Paul J. Valdes [6], Michel Crucifix [7] & Pedro N. DiNezio[1]

Much of the global annual mean temperature change over Quaternary glacial cycles can be attributed to slow ice sheet and greenhouse gas feedbacks, but analysis of the short-term response to orbital forcings has the potential to reveal key relationships in the climate system. In particular, obliquity and precession both produce highly seasonal temperature responses at high latitudes. Here, idealized single-forcing model experiments are used to quantify Earth's response to obliquity, precession, $CO_2$, and ice sheets, and a linear reconstruction methodology is used to compare these responses to long proxy records around the globe. This comparison reveals mismatches between the annual mean response to obliquity and precession in models versus the signals within Antarctic ice cores. Weighting the model-based reconstruction toward austral winter or spring reduces these discrepancies, providing evidence for a seasonal bias in ice cores.

[1] Institute for Geophysics, University of Texas at Austin, Austin, TX 78758, USA. [2] Department of Earth Sciences, University of Southern California, Los Angeles, CA 90089, USA. [3] School of Earth Science and Environmental Sustainability, Northern Arizona University, Flagstaff, AZ 86011, USA. [4] Department of Environmental Sciences and Institute for Earth, Ocean, and Atmospheric Sciences, Rutgers, The State University of New Jersey, New Brunswick, NJ 08901, USA. [5] Department of Earth Science, UC Santa Barbara, Santa Barbara, CA 93106, USA. [6] School of Geographical Sciences, University of Bristol, Bristol BS8 1SS, UK. [7] Earth and Life Institute, Université catholique de Louvain, Louvain-la-Neuve 1348, Belgium. Correspondence and requests for materials should be addressed to M.P.E. (email: michael.erb@nau.edu)

Slow temperature variations over Quaternary glacial-interglacial cycles have been dictated by changes in Earth's orbital obliquity, precession, and eccentricity, as well as slow feedbacks involving atmospheric composition and ice sheets. Although questions remain surrounding the interaction between orbital forcing and greenhouse gas and ice sheet dynamics[1,2], the present research focuses on the climate's direct short-term response to orbital forcing, which has been explored to varying degrees in past research[3–9]. Primarily, this paper focuses on the climate response to obliquity, or the axial tilt of the earth, which varies by up to ~2.5° with a period of 40 ka.

Changes in Earth's obliquity affect the latitudinal and seasonal distribution of insolation. Because this forcing is well known, it presents a good target to explore the climate system's response. Reduced obliquity results in positive annual mean insolation anomalies in the tropics while higher latitudes have increased insolation in winter and strongly decreased insolation in summer (see Fig. 1a in ref. [7]). These anomalies amplify the annual mean equator-to-pole insolation gradient and diminish the seasonal insolation cycle in both hemispheres. Unlike precession, obliquity's seasonally-varying insolation anomalies do not locally sum to zero in the annual mean.

In this paper, the climate response to obliquity is explored in model experiments and proxies. Model results show larger temperature responses at high latitudes than low latitudes, helped by radiative feedbacks and heat transport. In order to compare these modeled climate responses to signals preserved in proxy records, a linear reconstruction methodology is employed in which the results of the single-forcing model experiments are scaled by past climate forcings to produce a time-varying estimate of past climate change based on the model results. Results show that the model-based linear reconstruction has a larger direct obliquity response in Antarctica than suggested by Antarctic ice cores. The mismatch between the linear reconstruction and Antarctic ice cores is diminished when the linear reconstruction is averaged over austral winter or spring, offering evidence for a seasonal bias in the Antarctic ice cores. A seasonal bias in the Antarctic ice cores would not dramatically alter the recording of climate signals due to ice sheets and greenhouse gases—which make up the bulk of the climate response—but could alter the apparent magnitude and timing of climate responses to orbital forcing, which has implications for our understanding of past climate.

## Results

**Modeled temperature response to obliquity.** Two-coupled atmosphere-ocean general circulation models (GCMs) are used to model the climate response to changes in obliquity, precession, $CO_2$, and ice sheets. These are the Geophysical Fluid Dynamics Laboratory (GFDL) Climate Model 2.1 (CM2.1) and the National Center for Atmospheric Research (NCAR) Community Earth System Model (CESM) version 1.2. The temperature response to orbital forcing is also explored in the Hadley Centre Coupled Model version 3 (HadCM3). For CM2.1 and CESM, single-forcing equilibrium simulations are conducted at low (22.079°) and high (24.480°) obliquity, the extreme values of the past 900 ka[10]. All other forcings, such as greenhouse gases, ice sheets, precession, and eccentricity, are prescribed at preindustrial values. The models lack dynamic ice sheet and carbon cycle components, so orbital experiments capture the short-term climate response, but exclude slow feedbacks related to ice sheets and the carbon cycle. Additional single-forcing experiments isolate the climate response to changes in precession, $CO_2$, and ice sheets (Supplementary Table 1). The HadCM3 simulations[11], which are only used to examine the obliquity response, employ a different experimental design: simulations are conducted at regular intervals over the past 120 ka using different combinations of orbital forcings, and the obliquity response is extracted from the orbit-only simulations through statistical regression. In all models, obliquity is evaluated as a reduction in Earth's axial tilt by taking the difference between a low obliquity (22.079°) response and a high obliquity (24.480°) response.

In all three models, reduced obliquity produces pronounced cooling in the mid and high latitudes of both hemispheres and some areas of mild warming near the equator (Fig. 1). Obliquity redistributes insolation but does not alter the global average of annual mean insolation. The change in total radiative forcing, which accounts for albedo, is small (global mean radiative forcing anomalies are $+0.1\,W\,m^{-2}$ for CM2.1 and $+0.2\,W\,m^{-2}$ for CESM). Despite this mildly positive radiative forcing, cooling is much more widespread than warming, with all three models producing negative global mean temperature change. Global mean temperature anomalies are $-0.54\,°C$ for CM2.1, $-1.07\,°C$ for CESM, and $-0.48\,°C$ for HadCM3. The level of agreement between the models over most latitudes suggests that these results are fairly robust. The primary difference between models occurs in the Northern Hemisphere high latitudes, where CESM produces considerably more cooling than the other models, associated with widespread Arctic albedo anomalies in summer and fall due to changes in snow cover and sea ice.

Comparison of the obliquity signal in models and proxy records is central to the current analysis, so it is useful to understand what drives the modeled response. In the tropics, the temperature response is near zero even though annual mean insolation increases by $3.2\,W\,m^{-2}$ (radiative forcing increases by $2.4\,W\,m^{-2}$). Analysis of energy transports and radiative feedbacks helps explain the lack of a tropical temperature response to obliquity forcing. In part, the increased radiative forcing is balanced by increased poleward heat transport in both the atmosphere and ocean. In the atmosphere, the tropical portion of this modified heat transport is associated with changes in the Hadley circulation[6] (Supplementary Note 1 and Supplementary Fig. 1). From the extra-tropics to the poles, poleward heat transport is increased primarily through changes in sensible heat transport, associated with baroclinic eddies[6]. These atmospheric changes, together with increased ocean heat transport, redistribute heating and diminish the tropical warming. Some locations, especially parts of the tropical Atlantic and Pacific Oceans, even show a slight cooling (Fig. 1). Equatorial Pacific cooling in CM2.1 and CESM is greater below the surface, which has previously been suggested to explain tropical obliquity variations opposite to local insolation forcing in the early Pleistocene[12–14]. The present work shows that heat transport is enough to counter the direct insolation forcing at the surface, as already suggested in past work[15].

At higher latitudes, insolation is reduced during summer months. The change from high-to-low obliquity results in an annual mean insolation decrease of ~13 W m$^{-2}$ in the Arctic and Antarctic Circles–that is, a loss of 7% of the total local insolation. This translates to a mean radiative forcing of approximately $-6\,W\,m^{-2}$ within the Arctic Circle and $-5\,W\,m^{-2}$ within the Antarctic Circle in CM2.1 and CESM. Despite the increased equator-to-pole heat transport seen here and in another study[16], all three models show considerable cooling at high latitudes, facilitated by the effect of strong surface albedo and lapse rate feedbacks (Supplementary Note 2 and Supplementary Fig. 2). Low-latitude feedbacks are mixed, so these strong high-latitude feedbacks bring about global mean cooling in response to lowered obliquity, as explored in[4].

**Model/data comparison.** To determine how well the modeled responses are reflected in past climate, model results are

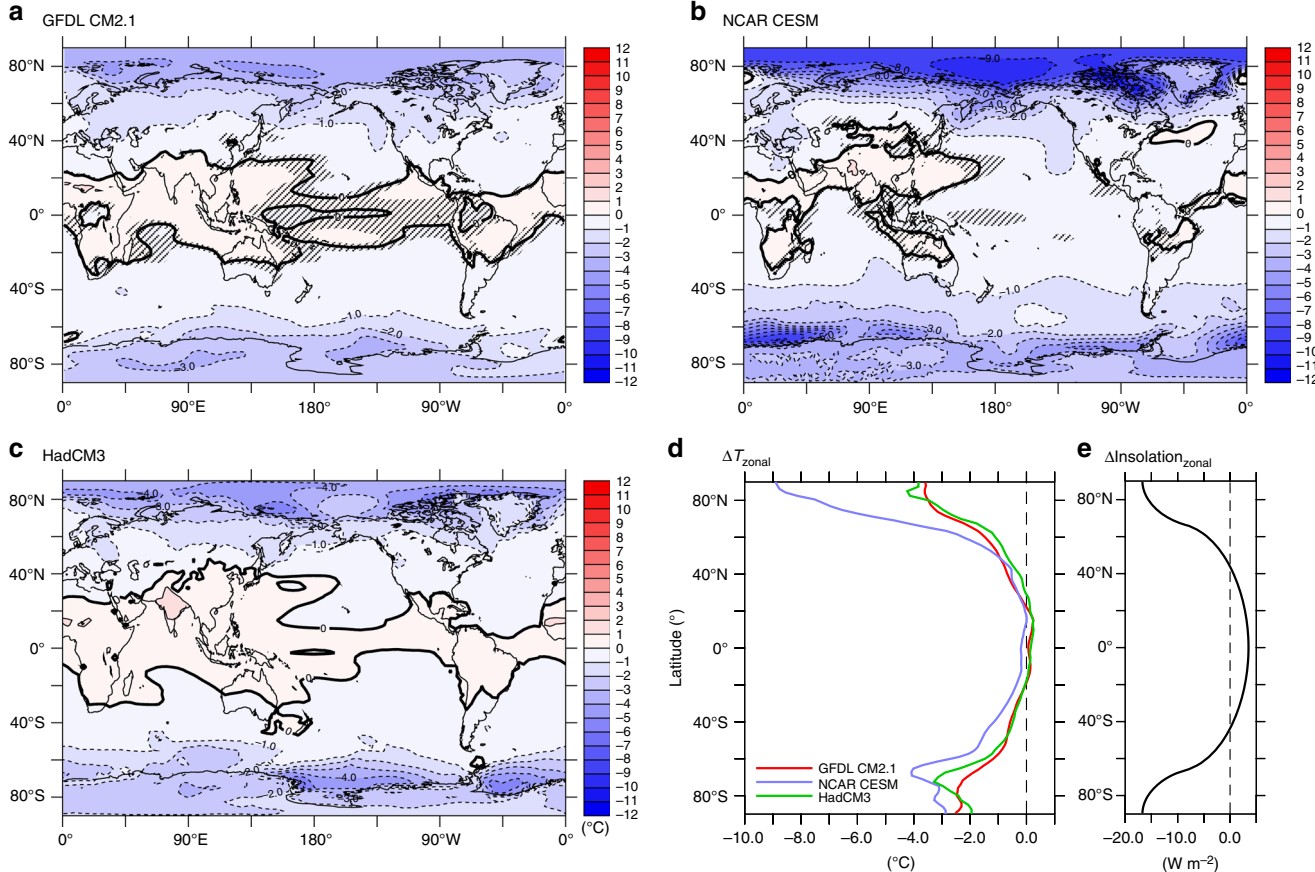

**Fig. 1** Modeled temperature response to obliquity. Annual mean surface air temperature anomaly (°C) for **a** CM2.1, **b** CESM, and **c** HadCM3 for the low–high obliquity experiments. Annual, zonal-mean anomalies for **d** surface air temperature and **e** insolation. In (**a**) and (**b**), hatching indicates anomalies which are not significant at a 0.05 level according to a two-tailed *t*-test (i.e., the means of the two 100-year model outputs are not found to be significantly different). Surface air temperature is calculated at 2 m for CM2.1 and CESM and 1.5 m for HadCM3. Global mean temperature anomalies are −0.54 °C for CM2.1, −1.07 °C for CESM, and −0.48 °C for HadCM3

compared with long proxy time series data from ice and ocean sediment cores. A linear reconstruction methodology, similar to that used in past work[17], is employed to compare the idealized single-forcing experiments with time-varying proxy records. Put simply, results from the single-forcing experiments, which isolate the climate response to obliquity, precession, $CO_2$, and ice sheets, are scaled by the time-varying forcings of the Late Quaternary to compute model-based estimates of the climate responses to individual forcings over the past several glacial cycles. Summed together, this creates a linear reconstruction of total climate change which can be compared to proxies. To put all records onto comparable age models, sediment cores were aligned to the LR04 benthic stack in past work[18], but some uncertainty exists in ice core age models. Leaving ice core records on their original published age models caused the timing of deglaciations to vary between records and the reconstruction, affecting results. To better align major climate features, Antarctic ice cores were aligned to the annual mean linear reconstructions. The timing of various records remains a potential source of uncertainty in the current analysis.

Past work suggests that the linear reconstruction methodology is more appropriate for some regions than others[17], with regions of sea ice presenting a particular difficulty. Still, because orbital forcing is well-characterized and produces highly seasonal responses over much of the world, focusing on regional orbital responses provide a valuable test of proxy records.

The total linear reconstruction covers 1–430 ka, the older bound of which is determined by the length of the shortest component used in the reconstruction. When interpreting the reconstruction, the obliquity and precession components should be regarded as the short-term climate responses to imposed insolation anomalies, while the additional contributions from greenhouse gas and ice sheet anomalies are quantified separately. This distinction between the direct short-term response, which quantifies the fast climate response to insolation changes, and the slower changes associated with ice sheet and carbon cycle feedbacks, should be kept in mind in the following analysis. See the methods section for more detail.

In general, the model-based linear reconstruction compares well with proxy temperatures from the European Project for Ice Coring in Antarctica (EPICA) Dome C[15], Dome Fuji[19], and Vostok[20] ice cores (Fig. 2). This linear reconstruction methodology allows the contribution from each forcing or feedback to be estimated. However, some of the larger mismatches between the ice cores and linear reconstructions appear to be related to the obliquity response. The linear reconstruction is generally too warm when obliquity causes Antarctic warming and too cold when obliquity causes Antarctic cooling. To explore this in a different way, a fitting methodology is employed: at each proxy location, the magnitude of the short-term obliquity component in the linear reconstruction is scaled to find the value that produces the best fit between the total linear reconstruction and the proxy

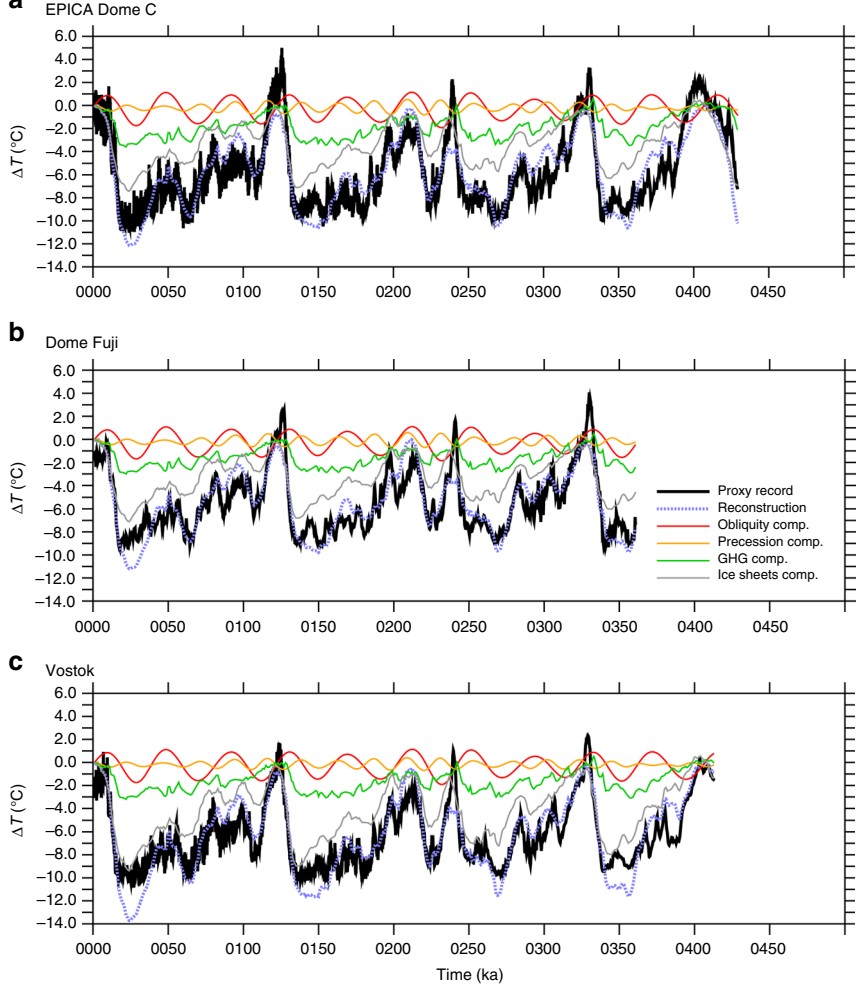

**Fig. 2** Temperature signals in ice cores and annual mean linear reconstructions. Temperatures over the past several glacial cycles from ice cores (black) and GFDL CM2.1 annual mean model-based linear reconstructions (dashed blue) for **a** EPICA Dome C, **b** Dome Fuji, and **c** Vostok. The model-based linear reconstruction is the sum of contributions from obliquity (red), precession (orange), greenhouse gases (green), and ice sheets (gray). Reconstruction $\Delta T$ is plotted relative to 1 ka and the proxy records are plotted with the same mean value as the reconstruction

record. This is done for the three Antarctic ice cores as well as a variety of records from other latitudes (Supplementary Table 2).

In general, outside of the Antarctic region the obliquity values supported by the proxy records are similar to the annual mean values of GCM simulations, or at least not different in a uniform way (Fig. 3, Supplementary Fig. 3). At low latitudes, the models and proxies both show relatively little sensitivity to obliquity, consistent with the hypothesis that changes in heat transport counteract much of the annual mean insolation forcing at those latitudes. At high latitudes, proxies show some obliquity signal[21,22], but there is a consistent mismatch in magnitude between the short-term obliquity response in models and that inferred from Antarctic proxies. Multiple long ice cores show similar responses in Antarctica, so the following analysis focuses on that region rather than Greenland. The Antarctic ice cores support only 24–33% of the fast, direct response to obliquity simulated by CM2.1 (Fig. 3), and 44–88% of the response in CESM (Supplementary Fig. 3). This discrepancy presents an intriguing question, and several possible hypotheses to explain the mismatch are outlined below.

One possible explanation involves the alignment of the proxy records. As stated earlier, the present analysis draws a distinction between the fast response to insolation forcing, which is captured in the orbital GCM simulations, and the longer-term obliquity response associated with slow feedbacks. If the age models are not correctly aligned, the comparison may suffer. The possibility is further explored using spectral analysis (Supplementary Note 3).

A second possible explanation for the mismatch is that the models may be too sensitive to obliquity forcing in the Antarctic. This could happen, for instance, if the modeled surface albedo or lapse rate feedbacks are too strong in that region, a negative feedback is absent, or if the southward heat transport from lower latitudes is too weak. The role of heat transport in dampening the Antarctic response to obliquity has been explored in past work using the LOVECLIM Earth system model of intermediate complexity and the MIROC GCM[16]. However, because the temperature response is generally robust across the three models explored in the present work (Fig. 1), any shortcoming would likely have to be systematic across these models. Additionally, the radiative forcing due to obliquity is large at high latitudes ($-5$ W m$^{-2}$ in the Antarctic Circle in CM2.1 and CESM), so a large temperature response seems reasonable, provided there is no error in the calculated strength of the orbital cycles. Dust is one possible source of error, as changes of dust[23] are not included in this methodology. Some GCM simulations suggest that dust has little effect on Antarctic temperatures[24,25]. However, dust may have

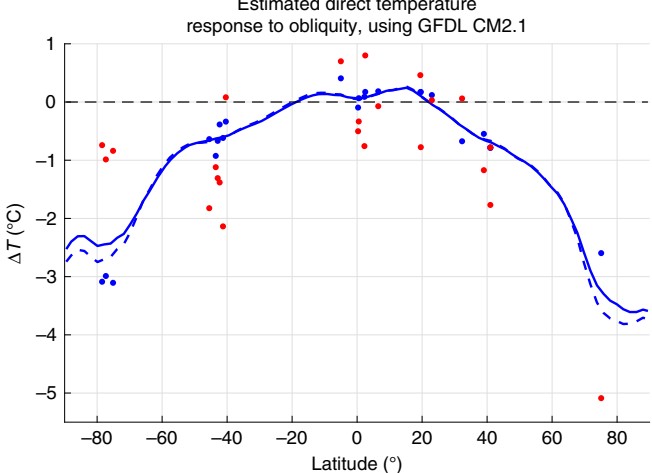

**Fig. 3** Obliquity temperature response in GFDL CM2.1 and estimated from proxies. Temperature changes at specific proxy locations in the GFDL CM2.1 low–high obliquity experiment (blue dots) and the apparent direct short-term obliquity response in proxies (red dots), which were derived by scaling the modeled obliquity response in the linear reconstructions to match the proxies. To show how the modeled temperature anomalies at proxy locations compare to zonal-mean values, lines show the zonal-mean anomalies for both surface air temperature (solid) and surface temperature (dashed) in GFDL CM2.1

considerable indirect effects on clouds[26], so the potential impact of dust in the present study is unclear. The mismatch could also stem from differences in the obliquity temperature response at the surface and higher in the atmosphere, where the snow preserved in ice cores forms, but the model results do not seem to support this. However, the discrete and course nature of the models' vertical grids makes this possibility difficult to evaluate.

An additional possible reason for the mismatch concerns shortcomings of the linear reconstruction methodology. Because the CM2.1 and CESM linear reconstructions are based on idealized single-forcing experiments, where one forcing is applied and all others are prescribed at preindustrial levels, any nonlinearities between multiple forcings are excluded. This may be important, for example, if the climate response to obliquity would be different under glacial boundary conditions, especially since glacials are more common than interglacials during the period of comparison, the Late Quaternary. Hypothetically, a diminished climate response to obliquity during colder conditions could help explain the mismatch. However, an analysis of the Antarctic temperature response to obliquity under a variety of background conditions conducted with a climate emulator using HadCM3 simulations[27] suggests that the climate response to obliquity is not heavily dependent on ice sheet extent or greenhouse gas concentrations (Supplementary Note 4 and Supplementary Figs. 11, 12).

**Possible seasonal bias in Antarctic ice cores**. Another possible explanation for the model–data mismatch involves the seasonal cycle. Obliquity affects high-latitude insolation primarily in the summer months, so the temperature response is larger in summer than in winter, which is subject to polar night. The obliquity signal could be diminished if, for example, the ice cores preferentially record winter more than summer temperature—the relatively small winter temperature anomalies would be preferentially recorded. A seasonal bias would affect the recorded precession

signal as well, potentially both in timing and magnitude, because of the seasonal nature of the temperature response to precession. The degree to which a seasonal bias can modify the recorded climate signal has been discussed in past work, and caution should be used when attempting to separate orbital signals which represent the actual climate response from those introduced by a seasonal bias in the recording mechanism[28]. It is worth noting that a seasonal bias would not strongly affect the greenhouse gas and ice sheet signals in Antarctic ice core records because, at least in the models, the temperature responses to ice sheets and greenhouse gases do not vary extensively throughout the year at the ice core sites (Supplementary Figs. 4, 5). Therefore, the temperature response to changes in greenhouse gases or ice sheets, which appear to be the bulk of the signal, should be recorded relatively accurately regardless of any potential seasonal bias, which is beneficial for studies of climate sensitivity. This point is relevant for lagged orbital responses as well. The EPICA Dome C record, for example, has a pronounced obliquity signal, but it lags the obliquity forcing by 5 ka[15], suggesting the influence of slow feedbacks like ice sheets. Dome Fuji also contains obliquity-scale lags associated with slow feedbacks[29]. Any temperature response due to these slow feedbacks should be captured in the ice cores, regardless of any potential seasonal bias, because temperature responses due to ice sheets and $CO_2$ are not very seasonal. Only the fast response to obliquity and precession (i.e., the temperature response to radiative forcing plus fast feedbacks) would be very distorted by a seasonal bias in the recording mechanism due to the large seasonality of these responses (Supplementary Fig. 5). Because of the largely seasonal nature of obliquity and precession signals, they provide a way to probe potential seasonal biases in long proxy records. In these ice core records, it is the fast, direct obliquity response which might be under-represented.

To calculate whether a seasonally weighted linear reconstruction would better match the proxy temperature records, mismatches are computed between each proxy record and seasonally weighted linear reconstructions (Fig. 4). A boxcar weighting function is employed to compute the mean over a continuous span expressed in months. Two factors are allowed to vary in this analysis: the center month of the mean and the time spanned by the mean, ranging from one to all twelve months. The general results do not differ substantially if a sinusoidal weighting function is used instead.

For CM2.1, linear reconstructions weighted toward Antarctic winter or spring best match the proxy records for all three Antarctic ice cores (Figs. 4, 5). On average, the best weightings reduce the root mean square error (RMSE) to 86% of their annual mean values. Some of the remaining mismatch is due to higher-frequency variability, which the reconstruction methodology is not expected to capture. When all time series are filtered in the obliquity band, the best weightings reduce the RMSE to 34% of the annual mean values, constituting a much better match for this component (Supplementary Figs. 6-8).

The improved fit is attributable to two factors. First, the local temperature response to obliquity is reduced in these months, reducing the obliquity signal. Second, this seasonal weighting alters the precession signal to generally be more consistent with precession-scale temperature variability in the proxy record, such as some of the variations in Marine Isotope Stage (MIS) 5a–e and 7 (Fig. 5, Supplementary Figs. 6-8). A seasonal mean can modify the apparent timing of these precession peaks because different months are warmed in different phases of the precessional cycle[30]. Conducting these calculations with CESM gives approximately the same results (Fig. 4d–f), with the best weighted reconstructions reducing RMSE to 91% of the annual mean values on average. When all time series are filtered in the

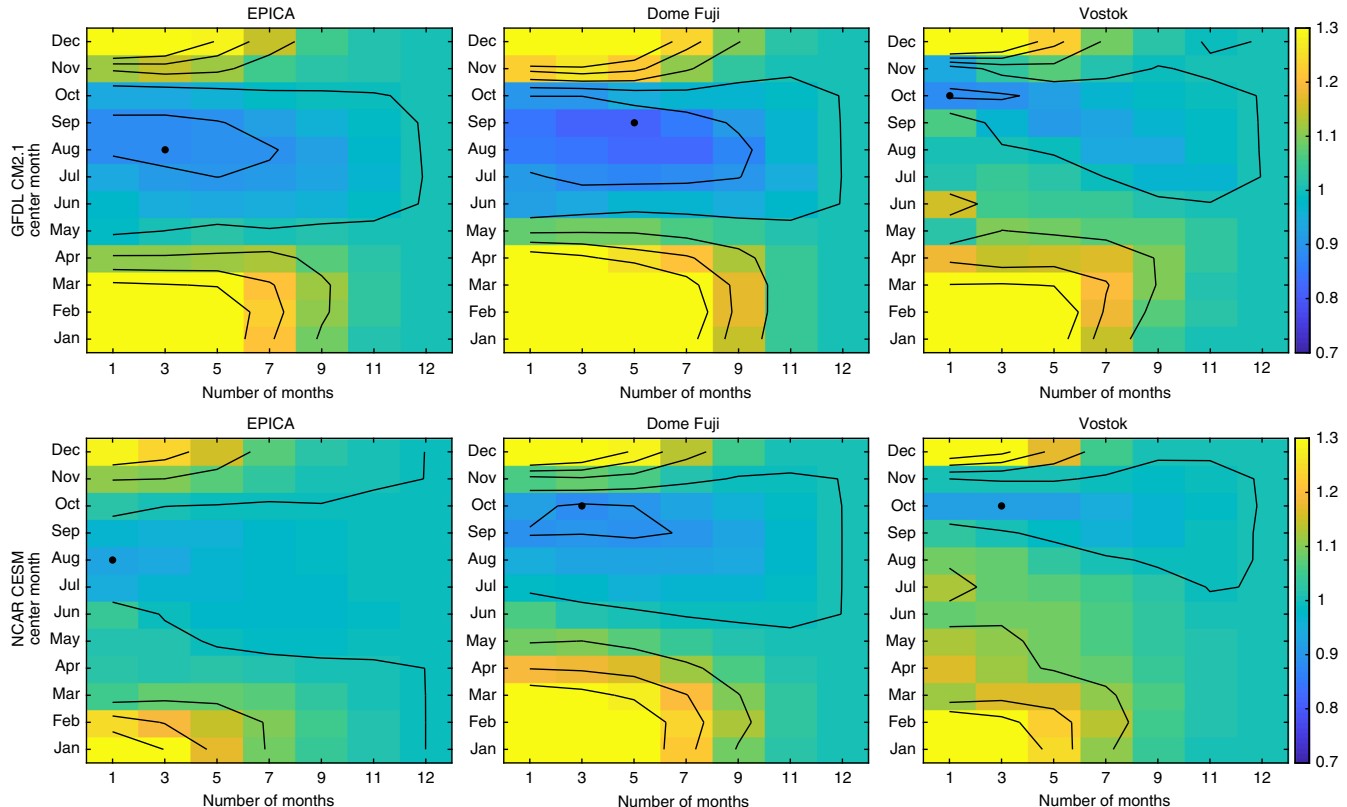

**Fig. 4** Mismatch between ice cores and linear reconstructions using different monthly averages. Relative mismatch between proxy temperature records and linear reconstructions using different monthly averages. Each grid box corresponds to the mismatch for a linear reconstruction averaged over a portion of the year. The y-axis defines the central month of the mean and the x-axis defines the number of months included in the mean. The rightmost column (labeled 12) indicates a 12-month mean. Values are the root mean square error (RMSE) of each monthly mean case normalized by the RMSE of the annual mean case, so monthly mean reconstructions which better match the proxy record compared to the annual mean case have values <1. Contours are drawn from 0.7 to 1.3 with a contour interval of 0.1. The black dot indicates the lowest RMSE. As an example of how to interpret this figure, the lowest RMSE in the top center panel corresponds to a 5-month average centered on September—in other words July to November. The calculation is done for all three Antarctic locations (EPICA Dome C, Dome Fuji, and Vostok) for (top) CM2.1 and (bottom) CESM

obliquity band, the best weightings reduce RMSE to 62% of the annual mean values for the CESM case.

A bias in Antarctic ice cores toward austral winter or spring is relatively consistent with some past studies. A comparison of temperature trends between EPICA and a transient simulation of the last 21 ka with ECBilt-Clio showed that the proxy record was best matched when the simulation was averaged over austral spring[31]. Another study showed that a seasonal bias could explain why the precession signal in Antarctic ice cores appears to be out of phase with local insolation[30]. That research focused on the timing of precession variability, and here we show that this explanation helps improve the fit of the obliquity signal as well. Performing the analysis by applying seasonal weighting to the obliquity or precession components alone shows that mismatch is generally reduced in both cases (Supplementary Figs. 9, 10). This provides two separate lines of evidence that Antarctic temperature records may be biased toward winter or spring. Previous work has suggested that the precession signal in Antarctic temperature may be a response to the duration of the summer, which is longer when perihelion occurs at the Southern Hemisphere winter solstice[32]. However, this explanation is undermined by the fact that such an orbital configuration produces annual mean cooling rather than warming over most of Antarctica in both the CM2.1 and CESM. Additionally, this explanation would not help explain the obliquity discrepancy.

A seasonal bias in ice cores could result from several factors. In particular, the mean isotopic signature could be biased by preferential snowfall during a particular part of the year. As compiled in previous work (Fig. 2a in ref. [30]), observations from Vostok, Mizuho, and Dome Fuji stations, as well as data from the European Centre for Medium-Range Weather Forecasts (ECMWF) suggest that Antarctic snowfall is reduced during Antarctic summer, in line with this explanation. This snowfall seasonality is supported by some modeling results[33], but not by others. In general, it is unclear whether models can simulate Antarctic precipitation accurately enough to be useful for this purpose[34]. The importance of seasonal precipitation to the interpretation of ice cores has been previously argued[35], as well as the potential importance of seasonal sublimation[30,36]. If precipitation is reduced during Antarctic winter instead, it would be evidence against the seasonality argument presented in this paper.

Considerable uncertainties still surround the topic of isotopic signatures in ice cores. In addition to seasonal variations, covariance between temperature and precipitation on synoptic timescales has the potential to affect isotopic ratios, which could be troublesome if these relationships change over time, although this effect may be less important for the interior of East Antarctica than for more coastal regions[37,38]. Other influences on isotopic composition could stem from changes in the temperature and source of evaporated moisture, regional changes

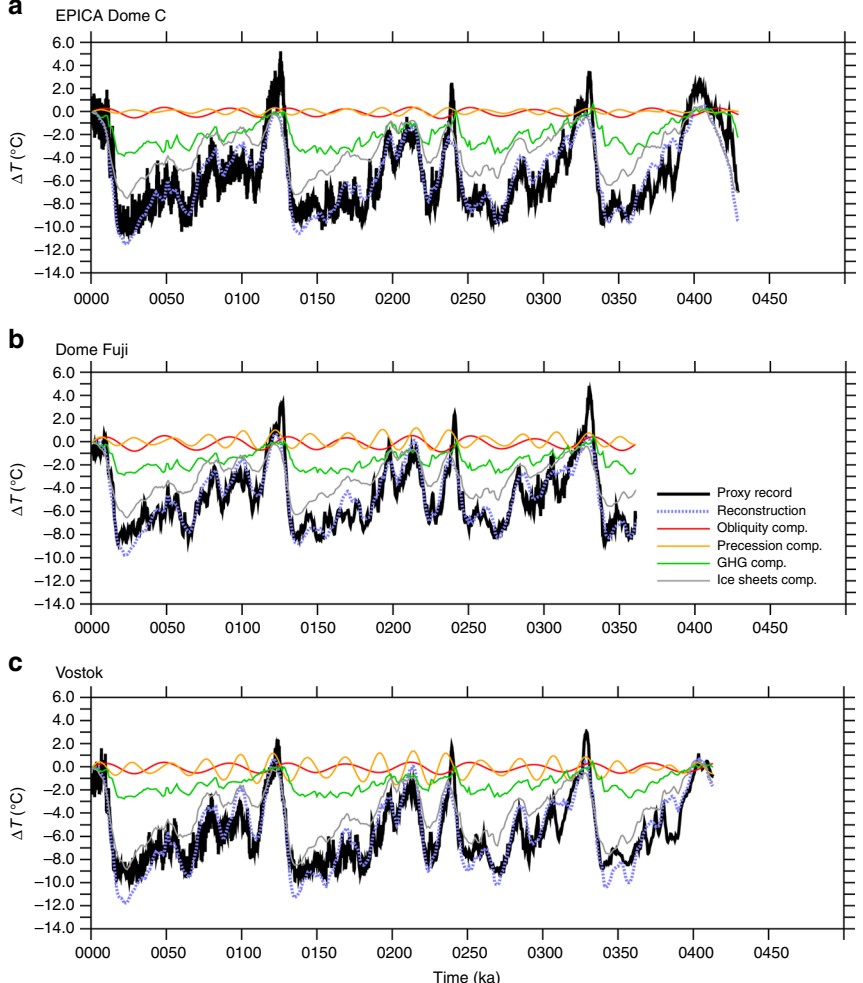

**Fig. 5** Temperature signals in ice cores and seasonally-averaged linear reconstructions. Like Fig. 2, but using the best fit seasonal weighting for the GFDL CM2.1 linear reconstruction (black dots in upper row of Fig. 4). Temperatures over the past several glacial cycles from ice cores (black) and GFDL CM2.1 seasonally-averaged model-based linear reconstructions (dashed blue) for **a** EPICA Dome C, **b** Dome Fuji, and **c** Vostok. The model-based linear reconstructions are the sum of seasonally weighted contributions from obliquity (red), precession (orange), greenhouse gases (green), and ice sheets (gray). The weighting is **a** July–September for EPICA Dome C, **b** July–November for Dome Fuji, and **c** October for Vostok, although other seasonal means also improve the fit compared to the annual mean case (Fig. 4). The seasonally weighted reconstructions shown here generally match the obliquity and precession-scale variability in the proxy records better than the annual mean reconstructions shown in Fig. 2

in ice sheet elevation, changes in lapse rate, and more[29,38–41]. Investigating all of these factors is beyond the scope of the present analysis. We focus on the potential impact of a seasonal bias because it is the most straightforward explanation for the apparent model/data temperature discrepancy presented here, but other explanations are certainly possible.

## Discussion

Our results suggest that a seasonal bias may exist in Antarctic ice cores. Such a bias would help explain the relative lack of in-phase obliquity signal in these records, as well as aspects of precession-scale variability. Conversely, this analysis strongly argues against a pronounced austral summer bias; such a bias would generally produce a larger obliquity signal and a precession signal out of phase with the precession-scale variability in the ice core records. If ice cores are recording an austral winter/spring seasonal signal, then the annual mean Antarctic temperature record has the potential to look somewhat different: obliquity-scale variability

should be somewhat larger and precession variability should generally be shifted a little in timing (cf. Figures 2 and 5). As discussed in previous work[30], a seasonal signal would reduce the need to explain precessional variability as a response to Northern Hemisphere summer insolation forcing[42], which has been difficult to justify, especially if Southern Hemisphere climate variations precede those in the Northern Hemisphere[32,43]. If Antarctic orbital temperature variability does respond more directly to local insolation, Southern Hemisphere insolation could have a larger role in producing past climate variations than typically assumed. Additionally, a seasonal bias should not greatly interfere with the use of Antarctic ice core records to estimate climate sensitivity, as the modeled Antarctic temperature responses to $CO_2$ and ice sheet changes are relatively consistent throughout the year (Supplementary Figs. 4, 5). An exception to this would be if the climate response to obliquity is used when calculating climate sensitivity. The general method outlined in this paper may be useful for exploring potential biases in other records, as long as the modeled responses are considered robust in the regions of investigation. As

more data becomes available, further exploration of these potential seasonal biases is encouraged.

## Methods

**Single-forcing GCM experiments.** To quantify the climate responses produced by changes in obliquity, precession, $CO_2$, and ice sheets, idealized single-forcing experiments were conducted, where only one forcing is applied while other forcings are prescribed at preindustrial levels. Simulations were conducted with the GFDL CM2.1 and the NCAR CESM1.2. Forcing parameters used for the simulations are given in Supplementary Table 1, and a broader overview of the CM2.1 simulations is presented in past work[17]. CESM simulations were generally forced with identical values to CM2.1 simulations except for slight differences in the control values. For ice sheet-forced simulations, GFDL CM2.1 uses the ICE-5G last glacial maximum (LGM) ice sheets[44] and NCAR CESM uses the Paleoclimate Modelling Inter-comparison Project Phase III (PMIP3) LGM ice sheets, which are a blend of three ice sheet reconstructions, with added ice shelves in the western Labrador Sea[45]. Simulations were run for 500 years or longer. These experiments are here referred to as 'fingerprint' experiments, since they quantify the distinct climate response patterns to individual forcings. A different experimental design was employed for the HadCM3 simulations used in Fig. 1c, d, as described in Supplementary Note 5.

For CM2.1 and CESM monthly output, an adjustment was typically made to the calendar. When the timing of perihelion is modified, as in some of the simulations described above, the speed at which Earth travels through different parts of its orbit is altered according to Kepler's second law. This means that calendar months in different simulations will correspond to different parts of Earth's orbit, complicating the interpretation of monthly anomalies. This 'calendar effect' has been described in past research[46]. To account for this complicating factor, all monthly output from CM2.1 and CESM were processed onto a common fixed-angular calendar, in which every 'month' corresponds to a 30° arc of Earth's orbit. This calculation was done using the method described in[47]. No monthly results from HadCM3 are presented, so no calendar correction is needed.

**Linear climate reconstructions.** To estimate the effects of each forcing in the past, and to allow for comparison with proxy time series data, a linear reconstruction methodology is employed. Model output from the single-forcing experiments is scaled by past forcings using an updated version of the method described in the appendix of[17]: anomalies from orbital experiments (consisting of simulations for high and low obliquity, simulations with precession at the two solstices and two equinoxes with increased eccentricity, and a simulation with zero eccentricity) are scaled by orbital parameters[48]; Anomalies from the $CO_2$ experiment are scaled by the radiative forcing of $CO_2$ from a combined $CO_2$ record from EPICA Dome C and Vostok[49] as well as the radiative forcing of $CH_4$ from EPICA Dome C[50], computed according to the equations in Table 6.2 of the Intergovernmental Panel on Climate Change (IPCC) Third Assessment Report[51]; Anomalies from the ice sheet experiment are scaled by sea level changes, which is used as an analog for ice sheet volume. The sea level record is a stack of seven different sea level records computed using different methods[52]. Potentially, uncertainties in the magnitude of obliquity variability in the sea level record could affect the results of this paper, especially if obliquity variations unrelated to sea level are also being recorded in the sea level record. However, the use of a multi-record stack rather than an individual sea level record should help reduce the impact of potential uncertainties in any particular method. It should also be noted that the climate response to ice sheets may relate to both ice sheet area and volume, but a linear scaling with sea level has been used here for simplicity. An additional source of uncertainty concerns the timing of individual ice sheet changes relative to the global mean sea level record. In the present work, the linear reconstruction assumes that all ice sheets vary synchronously with the global mean sea level record. However, evidence suggests that after the LGM, at least, Antarctic ice melt may have lagged considerably behind the global sea level changes[44]. Differences between the timing or characteristics of Antarctic ice sheet changes and global sea level anomalies could affect results of this analysis, especially considering that orbital-scale variations are present in the sea level curve (see Supplementary Note 3). Also note that the global mean sea level curve contains precession-scale variability approximately in line with Northern Hemisphere summer insolation; this means that the precession signal in the linear reconstruction is already largely in phase with the Antarctic ice core signals, even in the annual mean reconstruction (Supplementary Figs. 6-8). While it is unclear how realistic this is, the seasonal averaging of the linear reconstruction still generally shifts the timing of the direct precession signal and reduces mismatch in the current analysis. Furthermore, without quantitative knowledge of the ice sheet differences over the past few glacial periods, it is unclear how to account for this limitation in the present work. Finally, regarding greenhouse gases, by treating $CH_4$ as an equivalent amount of radiative forcing due to $CO_2$, the effect of $CH_4$ can be approximated without the need for a dedicated $CH_4$ simulation, which can be computationally expensive. The sea level stack was already aligned to LR04, and the other forcing time series ($CO_2$, $CH_4$, and orbit) are left on their published age models.

The computed 'linear reconstructions' estimate the contribution of individual forcings in producing past climate change, and can be summed together to compute a model-based estimate of total temperature change at any location. The linear reconstruction methodology used here has been updated from previous work[17] primarily in four ways: a slightly different data set of orbital parameters is used here (we use the same data set that was used to align the LR04 benthic stack)[48], greenhouse gas values from EPICA Dome C/Vostok are used instead of those from Vostok alone, a sea level stack is used rather than a single record, and linear reconstruction anomalies are computed relative to the preindustrial simulation rather than a zero eccentricity state.

**Comparing linear reconstructions with proxy records.** To explore climate signals preserved in proxies, long proxy temperature records were selected from multiple sources. For polar records, EPICA Dome C[15], Dome Fuji[19], Vostok[20], and NGRIP[53] were analyzed. For other regions, many proxy records which had been compiled for a global temperature stack[18] were used. The records in the temperature stack[18] were aligned to the LR04 benthic stack by that study's authors, generally using $\delta^{18}O$ to align records. The NGRIP ice core was left on its published ages[53], with a shift so that ages are relative to year 1950 instead of 2000. The Antarctic ice core records were aligned to the annual mean linear reconstructions, with these alignments done separately for the two models used in the analysis (GFDL CM2.1 and NCAR CESM). The alignment was done using matching software[54,55]. If no alignment is done, offsets in the timing of deglaciations affect the results. To test that the analysis is not overly sensitive to the details of this alignment, an alternate approach was also taken: no alignment was performed on the ice core age models, but periods of deglaciation prior to the most recent one were omitted from the analysis. This alternate analysis yields results similar to the main results presented in this paper, suggesting that the details of this alignment do not overly affect the results. Not all records were well suited to the present analysis, so two selection criteria were applied. Records that span multiple orbital cycles are preferred, so any records which contained less than 100 ka of overlap with the linear reconstructions were rejected. Additionally, some records exhibit temporal variability distinctly different from the typical 'sawtooth' shape observed in the greenhouse gas and sea level forcing records, potentially because of uncertainties in the proxy temperature relationship. To remove records with unusual variability as well as records with apparent age model mismatch, records which do not match the unfitted annual mean linear reconstruction of either model with at least a coefficient of efficiency of 0.5 were excluded. This selects records, which already match the linear reconstruction somewhat well, a qualification which should be kept in mind when evaluating results.

To conduct the model–data comparison, proxy records were compared to the linear reconstruction at the location of the proxy record, using modeled surface temperature for sediment cores and NGRIP and surface (2 m) air temperature for the Antarctic ice cores. To determine how much of the modeled annual mean obliquity signal was reflected in the proxy records (Fig. 3), the annual mean obliquity term in the reconstruction was multiplied by a scaling factor ranging from −1000 to 1000. This results in a set of possible linear temperature reconstructions differing only by the magnitude of the direct obliquity signal. After removing the mean difference between the reconstruction and proxy, the RMSE was calculated for each version of the linear temperature reconstruction. The solution with the smallest RMSE was deemed to be the solution best supported by the proxy record. This 'best' scaling factor was used to calculate the red dots in Fig. 3 and Supplementary Fig. 3. This comparison was performed for each proxy record separately.

To determine if a seasonal-weighted linear reconstruction better matched the Antarctic proxy records, RMSE was calculated between each proxy record and a set of linear temperature reconstructions computed using different monthly weightings. As opposed to the calculation described above, no aspects of the linear reconstruction were directly scaled to be larger or smaller. Instead, climate signals only changed as a result of different monthly averages. Averaging windows were centered on each month and spanned either 1, 3, 5, 7, 9, or 11 months, for a total of 72 monthly weighting combinations plus the 12-month mean (Fig. 4). A boxcar weighting function was employed so that months were either included in the mean or excluded, although the use of a sinusoidal weighting function gave relatively similar results. Before computing RMSE, the mean difference between the linear reconstruction and proxy temperature time series was removed. Results are computed as RMSE for each monthly mean case divided by RMSE of the annual mean case (Fig. 4). Values less than 1 indicate instances in which a seasonally weighted reconstruction better matches the proxy record.

**Code availability.** Code from this study is available from the corresponding author upon reasonable request.

**Data availability.** Model output from this study is available on Zenodo (zenodo.org) for both GFDL CM2.1 (doi:10.5281/zenodo.1194480) and NCAR CESM (doi:10.5281/zenodo.1194490) data. Annual and monthly climatologies are stored. Monthly data on a fixed-angular calendar is not stored on Zenodo, but is available from the corresponding author upon reasonable request.

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

## Acknowledgements

The authors would like to thank J. Shakun for his data set of proxy-derived temperatures aligned to LR04, V. Masson–Delmotte for suggestions about past ice core research, and F. Parrenin for the Vostok temperature reconstruction data. M.P.E. was supported by a postdoctoral fellowship from the University of Texas Institute for Geophysics and a grant from the Paleo Perspective on Climate Change program of the National Science Foundation (Grant ATM0902735), as well as support from the University of Southern California and Northern Arizona University. This work was partially funded under the State of Arizona Technology and Research Initiative Fund (TRIF), administered by the Arizona Board of Regents. The authors would also like to thank the NOAA Geophysical Fluid Dynamics Laboratory (GFDL) at Princeton, the National Center for Atmospheric Research (NCAR) and the Computational & Information Systems Lab (CISL), and the Hadley Center for computational resources to complete these simulations.

## Author contributions

M.P.E. conducted most of the CM2.1 and CESM simulations and conducted the bulk of the analysis. C.S.J. provided additional analysis and feedback, A.J.B. provided guidance for some model simulations and analysis, and D.W.L. provided insight about the choice of proxy records. P.J.V. ran the HadCM3 simulations, M.C. conducted analysis of climate emulator results, and P.N.D. ran the CESM ice sheets simulation and an additional preindustrial simulation.

**Additional information**

**Competing interests:** The authors declare no competing interests.

