## [Peer Review File · Nature Communications]

Reviewers' comments:

Reviewer #1 (Remarks to the Author):

Review of Model evidence for a seasonal bias in Antarctic ice cores
by Erb et al.,

Erb et al., use a range of single forcing experiments of three coupled climate models to predict the past temperature evolution of the Quaternary. The results are compared to long proxy records from the tropics (sediment cores) as well as from Antarctica (ice-cores). While there is a general good agreement between model based predictions and proxy reconstructions, the predicted obliquity component in Antarctica is considerably stronger than the one reconstructed from ice-cores. One explanation for this mismatch, which was previously proposed from independent evidence, is that the ice-cores are biased towards Austral winter or spring. Therefore, conditional on that the model predictions are reasonable; the study provides an evidence for a seasonal bias in Antarctic ice-cores.

The study is well written and provides an important twist on earlier results of a potential bias in Antarctic ice-cores, as it shows that both, the precession signal (as proposed earlier) as well as the obliquity signal (shown here) independently provide evidence for a seasonal bias in ice-cores. Further it demonstrates the implications of a seasonal bias (important for fast orbital response, not so important for the response on CO₂ and ice-sheet).

The critical point of the study is that the ice-core bias inference is conditional on that the model results are correct which is clearly not guaranteed given the present understanding of the past climate dynamics. On the other hand, the authors demonstrate the results on three climate models and show that the result is also expected given the energy balance. However, to strengthen the results, I suggest that the authors also try to include other (preferably high-latitude) high resolution temperature proxy records into their data model comparison.

Given the new findings, and the importance of ice-cores in paleoclimate research I feel that the results are well suited for Nature Communications after the major revision including more proxy records, (or if not, giving good reasons why this is not reasonable) and the minor revisions detailed below.

Major points:

1.) I suggest including more (marine) proxy records if possible. Of course they will have a coarser resolution than ice-cores, and might be also biased by seasonal recording but they should be good enough to decide between the model and the (annual) ice-cores in Figure 3. As a starting point, Shakun, J. D., D. W. Lea, L. E. Lisiecki, and M. E. Raymo (2015), An 800-kyr record of global surface ocean and implications for ice volume-temperature coupling, *Earth and Planetary Science Letters*, 426, 58-68, doi:10.1016/j.epsl.2015.05.042, on which David Lea, also an coauthor on this paper contributed presents 49 SST records. I would guess that at least a small subset of them might also have a high enough resolution and be long enough to fill the gap between tropics and Antarctica or the high northern latitudes. Showing that the model results are consistent no non-ice core records in a region where a strong obliquity response is modeled would greatly strengthen the conclusions of the paper of an ice-core bias.

2.) The method to determine the obliquity component in the proxy data (Line 511-519) seems unusual. In case that the model and proxy obliquity component would be out of phase (even

slightly) (for climate or proxy e.g. time-uncertainty reasons), this would bias the estimate towards underestimating the proxy obliquity component and thus lead to wrong conclusions. Thus if this method is kept, adding a (Supplementary) figure of the proxy and model time-series both filtered in the obliquity band would allow to rule this out and showing this for annual mean and after seasonal weighting, would also allow the reader to visually judge the improvement.

Minor points:

Around Line 68 (and Line 461): "(for CM2.1 simulations, see text and Table 1 in 11)". Given that the model simulations are a core part, I would prefer a self-contained paper, thus I propose to add an own table summarizing all experiments instead of just citing reference 11.

Line 112-114: I would guess that these numbers strongly depend on the latitude and would propose to include the analyzed latitude range in the text.

Line 147; Fig.3: I don't understand Fig.3. I see red dots (proxies) blue lines (model) and blue dots (model at proxy positions?)

Line 182:

The role of the seasonal cycle (and seasonal bias of proxy records) influencing the amplitude of the orbital component is often discussed in the literature e.g. Laepple, T., and G. Lohmann (2009), Seasonal cycle as template for climate variability on astronomical timescales, *Paleoceanography*, 24(4), PA4201 and Huybers, P., and C. Wunsch (2003), Rectification and precession signals in the climate system, *Geophysical Research Letters*, 30(19).

Line 214-215 / Discussion 259ff: There is a 19% reduction; where is the remaining 81%. I would suggest to add a discussion of the remaining discrepancy as maybe this will allow to learn about the models or proxy records.

Line 229: "Preforming " ->Performing?

Line 246: "and seasonal sublimation may be important as well". Add ref29 as this is also discussed there ("...the dependence of the seasonal accumulation on the summer insolation intensity, caused by the temperature dependence of the snow ablation")

Line 417:

"In (a) and (b), hatching indicates anomalies which are not significant at a 0.05 level according to a two-tailed test"

Please provide more detail what was tested (t-test between the distributions formed by the single years of the two forcing cases?, how many years are averaged)

Line 423: Which model is used here?

Line 432: "responses best supported by proxies" was confusing to me as it was first unclear if this is the same as "Temperature response to obliquity in CM2.1 (blue) and inferred from proxies (red)". I would propose to either omit it, or to write a short method sentence along the lines of: The temperature response inferred from proxies was derived by rescaling the simulated obliquity response to match the proxy records.

Figure 3: Can you include the model results using the optimal season. This would help the reader

to get an impression of the improvement and allow to discuss the remaining discrepancy.

Supplement:

The choice of the models in the figures seems arbitrary. Fig. S1: CM2.1
Fig. S2-S3: CESM, Figure S4: both.

Reviewer #2 (Remarks to the Author):

The paper "Model evidence for a seasonal bias in Antarctic ice cores" addresses an important issue. For decades now, Antarctic ice-core-based temperature reconstructions have been interpreted as annual mean temperatures. This fundamental misconception is usually explained by the high present-day spatial correlation between dD or $d18O$ proxy and annual mean temperatures (Dansgaard etc). What is very often ignored is the point that the correlation between the spatial isotope values and any seasonal mean would give similarly good correlations. Furthermore, good spatial correlations does not translate to good temporal correlations. Similar mistakes have been made in the paleoceanographic community by comparing spatial core top data with observed annual mean SSTs. As shown in Timmermann et al. (2014), *Paleoceanography*, this assumption is flawed.

Given the fact that this misunderstanding has also survived for a long time in the ice core community, I find it very refreshing to see it challenged in the Erb study. We wrote a long discussion on seasonal biases (mostly austral spring bias) in Antarctic ice cores in our 2009 paper (Timmermann et al. 2009, *J. Climate* - paper should be cited). The Laepple study was a second attempt, but their interpretation of was somewhat hard to follow.

The Erb et al. study is an interesting attempt to attack the problem from the obliquity side. The paper has a number of really strong points and several weaker ones.

The paper can be streamlined considerably. The discussion on heat transport changes is neither new, nor relevant for the seasonal bias discussion.

Here is a list of suggestions that would further improve the manuscript:

Start the discussion from original the dD , $d18O$ data, then mention how Jouzel and others derive the reconstructed temperatures. A number of assumptions is usually made in this step that should be explicitly stated. Each of these assumptions can be easily challenged. This would help to really point out what other potential factors (besides seasonal biases) could lead to the model/data mismatch and why the canonical ice-core interpretation is actually quite uncertain.

A fundamental assumption of the Erb paper is that there is no synergy between the individual forced responses in polar regions. I find this really hard to believe, in particular in more coastal areas, and I am not convinced that referring to personal communication is making this assumption more trustworthy. I would encourage the authors to make a stronger argument and describe and quantify in which regions this approximation of linearity actually holds. I thought that the HadCM3 model was run with all forcings and single forcings. It should be doable.

Instead of discussing in great length the heat transport changes in relation to obliquity forcing, which is really not adding anything to the paper, the authors could provide further evidence for their seasonal bias argument:

A) plot the mean seasonal cycle of precipitation $P(m)$ in the ERA40 and the CGCMs used, and possible changes due to forcings

B) calculate the annual mean covariance of precipitation and insolation $\langle P(m) Q(m, \text{year}) \rangle_m$ (see figure below showing the seasonal contribution to the precipitation-weighted insolation difference

11 ka and 21ka)

C) calculate the annual mean covariance of precipitation surface fluxes $\langle P(m) Q_{net}(m, year) \rangle_m$

D) calculate the precipitation-weighted temperature: $\langle P(m) T(m, year) \rangle_m$

These figures can be compared step by step with the ice-core data. This comparison will further elucidate what actually the main reason is for the spring seasonal biases (fixed season spring has the largest orbital forcing (Timmermann et al. (2009)) and it is also a season with comparably high snowfall - at least for present-day conditions).

The argument in the paper is mostly based on an analysis of the obliquity signal. I think the authors miss out on an important opportunity to link the inferences they make on obliquity to the precessional response.

The analysis of both forcings, and the mismatch between model amplitudes and ice-core amplitudes on both timescales would make for a much more compelling case. This requires revision of the manuscript.

I encourage the authors to further exploit the CM2.1 precessional runs in this regard.

I do not really understand, how the ice-cores can have a winter seasonal bias (stated in the abstract). There is no orbital forcing and precipitation is at its minimum. So the respective contribution should be the same in the model and the ice cores.

I remember, Laepple made a similar argument. This needs to be explained in more detail and in more intuitive terms.

Reviewer #3 (Remarks to the Author):

This manuscript presents linearized simulations of Earth's climate during the last ~500 kyr based on single forcing sensitivity experiments of obliquity, precession, CO₂ and ice sheets. Three coupled atmosphere-ocean GCMs are employed: GFDL-CM2.1, NCAR-CESM and HadCM3. The control experiment is typical of the pre-industrial climate. The influence of obliquity is particularly analyzed. The 500 kyr-long simulations are then compared with temperature records from three Antarctic ice cores: EPICA Dome C, Vostok and Dome Fuji. It is found that the simulations have stronger obliquity components than the observations. This discrepancy can be reconciled by assuming the isotopic records in Antarctic ice cores records preferentially the winter temperature.

This manuscript is generally well written and easy to read, with hypotheses clearly formulated. So I enjoy reading it and I have very little comments regarding the writing. However, its content is not really new and relies on strong hypotheses that are far from proven.

Not new:

- These sensitivity experiments were already published in Erb et al. (JC, 2015), although it was only based on one AO-GCM at that time (GFDL-CM2.1).

- Moreover, a similar study concluding that there is a bias in Antarctic ice core temperature record was published (ref. 29), although it was based on the timing of precession instead of the amplitude of obliquity.

Based on strong hypotheses:

- These 500 kyr-long simulations are based on a linear climate response hypothesis which is far from being proven in the current manuscript. The only argument I found is based on a reference to a personal communication to M. Crucifix (l. 179), which I find a bit light for a so central justification.

- As for sea level reconstructions, the record from the Red Sea is used. But what if this planktonic record has a temperature bias correlated to obliquity? This would completely alter the main

conclusion of the current manuscript and should be at least discussed.

- What is important for climate is not really ice volume but iced areas, and the conversion between volume and surface might not be linear.
- As explained in the method section, there is no calendar conversion for the HadCM3 simulations. The varying length of season is of primary importance for glacial cycles simulations. Not accounting for this mechanism makes the HadCM3 simulations almost useless in my opinion.
- it is also explained in "method" that CO₂, CH₄ and sea level are aligned to the LR04 stack. What is the rationale behind that? I would certainly not expect a zero phase between these different climate proxies! Especially, sea level is always delayed with respect to the other climate proxies.
- In terms of greenhouse gases, only CO₂ was taken into account. Why not considering CH₄ as a fifth forcing? Or is it implicitly accounted for in the CO₂ forcing as the "method" section seems to suggest? (in which case this should be clarified in the main text).

Minor clarification comments:

- l. 260: why such a bias would explain the lack of in-phase obliquity signal in Antarctic ice cores? I don't immediately see the link here. I reckon the delayed obliquity influence in Antarctic records is due to the delayed sea-level forcing.

All in all, this manuscript has some value but it relies on many unproven hypotheses so I am not sure it deserves a high-impact factor journal such as Nature Communications. I let the decision up to the editor.

Reviewers' comments:

Reviewer #1 (Remarks to the Author):

Review of Model evidence for a seasonal bias in Antarctic ice cores
by Erb et al.,

Erb et al., use a range of single forcing experiments of three coupled climate models to predict the past temperature evolution of the Quaternary. The results are compared to long proxy records from the tropics (sediment cores) as well as from Antarctica (ice-cores). While there is a general good agreement between model based predictions and proxy reconstructions, the predicted obliquity component in Antarctica is considerably stronger than the one reconstructed from ice-cores. One explanation for this mismatch, which was previously proposed from independent evidence, is that the ice-cores are biased towards Austral winter or spring. Therefore, conditional on that the model predictions are reasonable; the study provides an evidence for a seasonal bias in Antarctic ice-cores.

The study is well written and provides an important twist on earlier results of a potential bias in Antarctic ice-cores, as it shows that both, the precession signal (as proposed earlier) as well as the obliquity signal (shown here) independently provide evidence for a seasonal bias in ice-cores. Further it demonstrates the implications of a seasonal bias (important for fast orbital response, not so important for the response on CO₂ and ice-sheet).

The critical point of the study is that the ice-core bias inference is conditional on that the model results are correct which is clearly not guaranteed given the present understanding of the past climate dynamics. On the other hand, the authors demonstrate the results on three climate models and show that the result is also expected given the energy balance. However, to strengthen the results, I suggest that the authors also try to include other (preferably high-latitude) high resolution temperature proxy records into their data model comparison.

Given the new findings, and the importance of ice-cores in paleoclimate research I feel that the results are well suited for Nature Communications after the major revision including more proxy records, (or if not, giving good reasons why this is not reasonable) and the minor revisions detailed below.

Major points:

1.) I suggest including more (marine) proxy records if possible. Of course they will have a coarser resolution than ice-cores, and might be also biased by seasonal recording but they should be good enough to decide between the model and the (annual) ice-cores in Figure 3. As a starting point, Shakun, J. D., D. W. Lea, L. E. Lisiecki, and M. E. Raymo (2015), An 800-kyr record of global surface ocean and implications for ice volume-temperature coupling, *Earth and Planetary Science Letters*, 426, 58-68, doi:10.1016/j.epsl.2015.05.042, on which David Lea, also an coauthor on this paper contributed presents 49 SST records. I would guess that at least a small subset of them might also have a high enough resolution and be long enough to fill the gap between tropics and Antarctica or the high northern latitudes. Showing that the model results are consistent no non-ice core records in a region

where a strong obliquity response is modeled would greatly strengthen the conclusions of the paper of an ice-core bias.

I agree. I emailed Jeremy Shakun and he sent me the proxy temperature records used in Shakun et al. 2015. Not all of these records are ideal for the current analysis, so I filter them in two ways: I removed records which had less than 100ka of data (since the present method benefits from longer records) and I removed records which did not have a coefficient of efficiency (CE) value of at least 0.5 with the unfitted reconstruction of either model. This second condition was applied because the current methodology works best with records which have the traditional “sawtooth” shape shared by the global-mean CO₂ and sea level curves. Records which do not share this shape may be affected by local temperature anomalies, and are therefore less suitable for this analysis. In addition to the Shakun et al. 2015 records, I tested some additional records from online repositories, but decided not to use them in the paper. These Shakun records have been added to Figs. 3 and S3, and a table detailing all of the records has been added to the supplemental materials (Table S2). These records show a more diverse array of climate responses, and there are certainly disagreements between the records and the annual-mean reconstructions. However, these new records do not show the same level of consistent disagreement with the model results as seen in the Antarctic ice cores.

2.) The method to determine the obliquity component in the proxy data (Line 511-519) seems unusual. In case that the model and proxy obliquity component would be out of phase (even slightly) (for climate or proxy e.g. time-uncertainty reasons), this would bias the estimate towards underestimating the proxy obliquity component and thus lead to wrong conclusions. Thus if this method is kept, adding a (Supplementary) figure of the proxy and model time-series both filtered in the obliquity band would allow to rule this out and showing this for annual mean and after seasonal weighting, would also allow the reader to visually judge the improvement.

To address this question, I have filtered the records in the obliquity and precession bands and conducted a spectral analysis. Text has been added to the supplemental materials and figures S6-8 show the three Antarctic records filtered in the obliquity and precession bands. These figures show that, even if timing is ignored, the obliquity signal in the annual-mean linear reconstruction is generally too large compared to the proxy records. These signals are better fitted by the seasonally-weighted linear reconstruction, suggesting a possible seasonal bias in the ice cores.

It should be pointed out that both the obliquity and sea level components of the linear reconstruction have a large obliquity signal. However, the obliquity component in the sea level record is lagged from the direct obliquity signal by a clear amount (~10 ka). The obliquity signal was left on its original (calculated) time axis, while the sea level record and all proxies (except NGRIP, which has a higher temporal resolution) are aligned to the LR04 benthic stack. Age uncertainties do remain a concern, and alternate ways of dating the ice cores are mentioned in the text. In the primary analysis, the ice cores are aligned to the annual-mean linear reconstructions. If the ice cores are left on their original age models, periods of rapid temperature changes during deglaciations do not line up, affecting the analysis (the analysis computes RMSE, so large errors at deglaciations affects results a lot). However, if deglaciation are ignored in the analysis, the results are again similar to those discussed in the paper.

Minor points:

Around Line 68 (and Line 461): "(for CM2.1 simulations, see text and Table 1 in 11)". Given that the

model simulations are a core part, I would prefer a self-contained paper, thus I propose to add an own table summarizing all experiments instead of just citing reference 11.

I have added a table with these forcing parameters to the supplementary materials: Table S1.

Line 112-114: I would guess that these numbers strongly depend on the latitude and would propose to include the analyzed latitude range in the text.

These values are calculated within the Arctic and Antarctic Circles (i.e. poleward of 66.5 degrees). I have modified the wording to make this clearer.

Line 147; Fig.3: I don't understand Fig.3. I see red dots (proxies) blue lines (model) and blue dots (model at proxy positions?)

The blue lines are zonal-mean model output, while the blue dots are model output at the specific proxy locations. Red dots are the values that best fit the proxies. I have changed the wording the Fig. 3 caption to make this clearer.

Line 182:

The role of the seasonal cycle (and seasonal bias of proxy records) influencing the amplitude of the orbital component is often discussed in the literature e.g. Laepple, T., and G. Lohmann (2009), Seasonal cycle as template for climate variability on astronomical timescales, *Paleoceanography*, 24(4), PA4201 and Huybers, P., and C. Wunsch (2003), Rectification and precession signals in the climate system, *Geophysical Research Letters*, 30(19).

I have added a reference to Huybers and Wunsch 2003. Laepple and Lohmann 2009 primarily discusses the rectification of seasonal forcings (i.e. the climate can respond unequally to equally-size insolation forcings in different seasons), which is not as relevant to this study. In this paper, we do not discuss rectification of seasonal signals, but instead the possibility that ice cores are preferentially recording a particular season rather than the annual-mean.

Line 214-215 / Discussion 259ff: There is a 19% reduction; where is the remaining 81%. I would suggest to add a discussion of the remaining discrepancy as maybe this will allow to learn about the models or proxy records.

Some of the remaining discrepancy is due to higher-frequency variability that this method is not expected to capture. I have added a sentence about this to the paper, and added the RMSE change when the records are filtered in the obliquity band. Additional discrepancy may be due to limitations of the linear reconstruction methodology or other factors. Some discrepancy, such as the magnitude of warmth during previous interglacials, is intriguing, but the reason for it is unclear.

Line 229: "Preforming " ->Performing?

Thanks. I've fixed it.

Line 246: "and seasonal sublimation may be important as well". Add ref29 as this is also discussed there ("...the dependence of the seasonal accumulation on the summer insolation intensity, caused by the temperature dependence of the snow ablation")

I've added it.

Line 417:

"In (a) and (b), hatching indicates anomalies which are not significant at a 0.05 level according to a two-tailed test"

Please provide more detail what was tested (t-test between the distributions formed by the single years of the two forcing cases?, how many years are averaged)

The t-test was computed to test the difference between the means of two 100-year model output. I have added some text in the Fig. 1 caption to make this clearer. The test was done according to the "Two-Sample t-Test for Equal Means" at the following page: <http://www.itl.nist.gov/div898/handbook/eda/section3/eda353.htm>

Line 423: Which model is used here?

Both Fig. 2 and Fig. 5 show results from the GFDL CM2.1. I've now specified this in the figure captions.

Line 432: "responses best supported by proxies" was confusing to me as it was first unclear if this is the same as "Temperature response to obliquity in CM2.1 (blue) and inferred from proxies (red)". I would propose to either omit it, or to write a short method sentence along the lines of:

The temperature response inferred from proxies was derived by rescaling the simulated obliquity response to match the proxy records.

I have changed the wording of the Fig. 3 caption to make this clearer. A fuller explanation is given in the methods section.

Figure 3: Can you include the model results using the optimal season. This would help the reader to get an impression of the improvement and allow to discuss the remaining discrepancy.

I think that this would make the figure too visually busy and more difficult to interpret. I have instead added a couple sentences to the text to mention the root mean square error reduction in the reconstruction after filtering the time series in the obliquity band.

Supplement:

The choice of the models in the figures seems arbitrary. Fig. S1: CM2.1

Fig. S2-S3: CESM, Figure S4: both.

Most of the supplemental figures show results from CM2.1, which is the primary model shown in the main paper. Figs. S2-S3 showed CESM because the variables necessary to compute all of the heat fluxes were not saved for GFDL CM2.1 (however, Figs. S2-3 have now been removed to streamline the paper's focus). Fig. 4 shows both to show how the feedbacks compare between models. Fig. S5 (now Fig. S3) is the CESM version of Fig. 3 in the paper.

Reviewer #2 (Remarks to the Author):

The paper "Model evidence for a seasonal bias in Antarctic ice cores" addresses an important issue. For

decades now, Antarctic ice-core-based temperature reconstructions have been interpreted as annual mean temperatures. This fundamental misconception is usually explained by the high present-day spatial correlation between δD or $\delta^{18}O$ proxy and annual mean temperatures (Dansgaard etc). What is very often ignored is the point that the correlation between the spatial isotope values and any seasonal mean would give similarly good correlations. Furthermore, good spatial correlations does not translate to good temporal correlations. Similar mistakes have been made in the paleoceanographic community by comparing spatial core top data with observed annual mean SSTs. As shown in Timmermann et al. (2014), *Paleoceanography*, this assumption is flawed.

Given the fact that this misunderstanding has also survived for a long time in the ice core community, I find it very refreshing to see it challenged in the Erb study. We wrote a long discussion on seasonal biases (mostly austral spring bias) in Antarctic ice cores in our 2009 paper (Timmermann et al. 2009, *J. Climate* - paper should be cited). The Laepple study was a second attempt, but their interpretation of was somewhat hard to follow.

I have included a citation of the Timmermann et al. 2009 paper.

The Erb et al. study is an interesting attempt to attack the problem from the obliquity side. The paper has a number of really strong points and several weaker ones.

The paper can be streamlined considerably. The discussion on heat transport changes is neither new, nor relevant for the seasonal bias discussion.

Here is a list of suggestions that would further improve the manuscript:

Start the discussion from original the δD , $\delta^{18}O$ data, then mention how Jouzel and others derive the reconstructed temperatures. A number of assumptions is usually made in this step that should be explicitly stated. Each of these assumptions can be easily challenged. This would help to really point out what other potential factors (besides seasonal biases) could lead to the model/data mismatch and why the canonical ice-core interpretation is actually quite uncertain.

A paper formatted in that way would be useful, but a full catalog of possible errors and misinterpretation of ice core proxies is beyond the scope of the current paper, and is an area of ongoing and uncertain research. The focus of this paper is, instead, to explore a possible explanation for the mismatch between the temperature records as they currently are and the model results. Other sources of error and disagreement are mentioned, but this is not intended to be a comprehensive review of the subject.

A fundamental assumption of the Erb paper is that there is no synergy between the individual forced responses in polar regions. I find this really hard to believe, in particular in more coastal areas, and I am not convinced that referring to personal communication is making this assumption more trustworthy. I would encourage the authors to make a stronger argument and describe and quantify in which regions this approximation of linearity actually holds. I thought that the HadCM3 model was run with all forcings and single forcings. It should be doable.

To address the topic of linearity, I have expanded the discussion of the climate “emulator” results of M. Crucifix in the supplemental materials. Two figures of these results have been added to the supplementary materials to show that, at least in the HadCM3 model, the assumption of linearity

appears to hold up quite well. M. Crucifix has been added as a coauthor for these additions. Regarding the coastal areas, it is possible that some nonlinearities have not been captured, but the three ice cores explored in this paper are not close to the coasts. The primary set of HadCM3 simulations discussed in the paper were run with certain sets of forcings, but not enough to make an analysis of linearity and nonlinearity possible. Still, the discussion added to the supplemental materials offers support for the use of the linear approach.

Instead of discussing in great length the heat transport changes in relation to obliquity forcing, which is really not adding anything to the paper, the authors could provide further evidence for their seasonal bias argument:

- A) plot the mean seasonal cycle of precipitation $P(m)$ in the ERA40 and the CGCMs used, and possible changes due to forcings
- B) calculate the annual mean covariance of precipitation and insolation m (see figure below showing the seasonal contribution to the precipitation-weighted insolation difference 11 ka and 21ka)
- C) calculate the annual mean covariance of precipitation surface fluxes m
- D) calculate the precipitation-weighted temperature: m

These figures can be compared step by step with the ice-core data. This comparison will further elucidate what actually the main reason is for the spring seasonal biases (fixed season spring has the largest orbital forcing (Timmermann et al. (2009)) and it is also a season with comparably high snowfall - at least for present-day conditions).

While the heat transport discussion is a little tangential to the discussion of possible seasonal biases in ice cores, it does help explain the modeled climate response to obliquity. The mismatch between the obliquity response in the model and that implied by the ice cores is central to the discussion of seasonal biases, so this is relevant to the paper. This discussion also highlights another interesting results of this work, which is the near-zero modeled response of tropical temperatures to obliquity forcing. Still, some of this discussion has been moved to the supplemental materials. Additionally, two figures showing calculated heat transports (Figs. S2-3 of the original draft) have been removed to improve the focus of the paper.

A deeper analysis of modeled precipitation changes is a worthwhile pursuit, but many GCMs have considerable trouble simulating the precipitation cycle over Antarctica. The extent to which models can accurately simulation Antarctic precipitation is argued in ¹, with Fig. 1 of the reply showing considerable mismatch between modeled and observed precipitation. Because the seasonality of precipitation may be poorly modeled, a more in-depth analysis of precipitation changes does not seem like it would lead to meaningful conclusions. The Sime response is cited in our paper to explain why a deeper analysis of modeled precipitation may not be useful. In general, GCMs are better at capturing temperature changes than precipitation changes, so this paper focuses its analysis on temperature.

The argument in the paper is mostly based on an analysis of the obliquity signal. I think the authors miss out on an important opportunity to link the inferences they make on obliquity to the precessional response.

The possible effect of a seasonal bias on the precession signal is already mentioned in the paper. See, for example, the fourth paragraph in the section titled "possible seasonal bias in Antarctic ice cores". Precession is also discussed to a greater extent in Laepple et al. 2011, which is cited.

The analysis of both forcings, and the mismatch between model amplitudes and ice-core amplitudes on both timescales would make for a much more compelling case. This requires revision of the manuscript.

Obliquity and precession signals are both taken into account in the current analysis, as shown by the seasonal fitting shown in Fig. 4. The seasonal weighting in Fig. 4 affects all components: obliquity, precession, ice sheets, and CO2. Additionally, to determine if this result is solely because of the obliquity or precession components, the analysis is repeated in cases where only obliquity or precession are affected by the seasonal weighting (Figs. S8 and S9). Both cases suggest that mismatch is reduced if the ice core is preferentially recording a local winter or spring temperature signal. Filtering the records in the obliquity and precession bands (Figs. S6-S8) also explore the mismatches for obliquity and precession separately.

I encourage the authors to further exploit the CM2.1 precessional runs in this regard.

I do not really understand, how the ice-cores can have a winter seasonal bias (stated in the abstract). There is no orbital forcing and precipitation is at its minimum. So the respective contribution should be the same in the model and the ice cores.

I remember, Laepple made a similar argument. This needs to be explained in more detail and in more intuitive terms.

As you suggest, obliquity has a smaller impact on winter temperatures than temperatures during the rest of the year (Figs. S9-10). However, because the Antarctic ice cores also show a smaller short-term obliquity signal than modeled for the annual mean, this is in line with a winter or spring seasonal bias. Regarding precipitation, there is some debate about the seasonality of Antarctic precipitation. Because of the uncertainties about this, I have not included precipitation in my argument. If Antarctic winter precipitation is indeed reduced compared to other seasons during the Quaternary, this may be evidence against the hypothesis presented in the paper. I have added a little extra text in the paper to make note of this.

Reviewer #3 (Remarks to the Author):

This manuscript presents linearized simulations of Earth's climate during the last ~500 kyr based on single forcing sensitivity experiments of obliquity, precession, CO2 and ice sheets. Three coupled atmosphere-ocean GCMs are employed: GFDL-CM2.1, NCAR-CESM and HadCM3. The control experiment is typical of the pre-industrial climate. The influence of obliquity is particularly analyzed. The 500 kyr-long simulations are then compared with temperature records from three Antarctic ice cores: EPICA Dome C, Vostok and Dome Fuji. It is found that the simulations have stronger obliquity components than the observations. This discrepancy can be reconciled by assuming the isotopic records in Antarctic ice cores records preferentially the winter temperature.

This manuscript is generally well written and easy to read, with hypotheses clearly formulated. So I enjoy reading it and I have very little comments regarding the writing. However, its content is not really new and relies on strong hypotheses that are far from proven.

Not new:

- These sensitivity experiments were already published in Erb et al. (JC, 2015), although it was only based on one AO-GCM at that time (GFDL-CM2.1).

- Moreover, a similar study concluding that there is a bias in Antarctic ice core temperature record was published (ref. 29), although it was based on the timing of precession instead of the amplitude of obliquity.

The current paper complements and expands upon those previously published papers.

Based on strong hypotheses:

- These 500 kyr-long simulations are based on a linear climate response hypothesis which is far from being proven in the current manuscript. The only argument I found is based on a reference to a personal communication to M. Crucifix (l. 179), which I find a bit light for a so central justification.

I have expanded the discussion of linearity in the supplemental materials. Instead of a personal communication, two figures have been added to the supplementary materials to show results from a "climate emulator" experiment with the HadCM3 (Figs. S11 and S12). This climate emulator was constructed by M. Crucifix (who has been added as a coauthor) by running HadCM3 simulations with a wide variety of forcing combinations, then using a technique to estimate the climate response over the full, continuous forcing space. Fig. S12 shows that the mean Antarctic temperature response to obliquity in HadCM3 is approximately the same (and linear) regardless of the state of ice sheets or CO₂, supporting the linear approach employed in the paper.

- As for sea level reconstructions, the record from the Red Sea is used. But what if this planktonic record has a temperature bias correlated to obliquity? This would completely alter the main conclusion of the current manuscript and should be at least discussed.

I have replaced the sea level record from the Red Sea with a sea level stack computed from seven different sea level records². While each of the records could still have biases, the use of a stack should offset this to some extent. Using this stack, the conclusions of the paper remain largely the same.

- What is important for climate is not really ice volume but iced areas, and the conversion between volume and surface might not be linear.

I have added a sentence mentioning this to the methods section.

- As explained in the method section, there is no calendar conversion for the HadCM3 simulations. The varying length of season is of primary importance for glacial cycles simulations. Not accounting for this mechanism makes the HadCM3 simulations almost useless in my opinion.

I have removed the majority of the HadCM3 analysis, both because of the lack of calendar conversion and other shortcomings of those simulations.

- it is also explained in "method" that CO₂, CH₄ and sea level are aligned to the LR04 stack. What is the rationale behind that? I would certainly not expect a zero phase between these different climate proxies! Especially, sea level is always delayed with respect to the other climate proxies.

The temporal alignment used in this paper has been modified. All sediment cores were aligned to the LR04 benthic stack by Shakun et al. 2015, generally using $\delta^{18}O$, by J. Shakun. For the ice cores, we align those to the annual-mean linear reconstructions themselves using Lorraine Lisiecki's Match program. This was done because, when left on their original age models, clear differences exist in the

timing of deglaciations, which affects the analysis in the paper. Sea level was already aligned to LR04, and the CO₂ and CH₄ forcings are left on their original ages. Using this new method, we get similar results as the original analysis. A sensitivity test was also made: the analysis was done with no adjustment to the ice core ages, but periods of deglaciation were ignored in the analysis. This alternate method produce similar results. Additional explanation about this has been added to the paper's methods section.

- In term of greenhouse gases, only CO₂ was taken into account. Why not considering CH₄ as a fifth forcing? Or is it implicitly accounted for in the CO₂ forcing as the "method" section seems to suggest? (in which case this should be clarified in the main text).

CH₄ is accounted for. No dedicated CH₄ experiment was performed, but CH₄ is included according to its radiative forcing, using the CO₂ simulation to approximate the response. I have added a sentence to the methods section to make this clearer.

Minor clarification comments:

- l. 260: why such a bias would explain the lack of in-phase obliquity signal in Antarctic ice cores? I don't immediately see the link here. I reckon the delayed obliquity influence in Antarctic records is due to the delayed sea-level forcing.

The obliquity simulations show that, even without ice sheet and sea level feedbacks, obliquity is modeled to produce pronounced temperature change at the high latitudes. This is the temperature response that should happen immediately, without waiting for slow ice sheet or greenhouse gas feedbacks, and this direct response to insolation change would be expected to appear in ice core record as well. Since the direct response in ice cores appears to be much smaller, the paper explores potential reasons for this discrepancy. Because some seasons respond to obliquity less strongly than others, a seasonal bias in the ice cores could explain the apparent discrepancy between the models and data. I have a sentence in the "model/data comparison" section to clarify this.

All in all, this manuscript has some value but it relies on many unproven hypotheses so I am not sure it deserves a high-impact factor journal such as Nature Communications. I let the decision up to the editor.

In addition to the changes discussed above, other changes have been made to improve the writing or analysis in the paper.

References

1. Sime, L. C. & Wolff, E. W. Antarctic accumulation seasonality. *Nature* 479, E1-2-4 (2011).
2. Spratt, R. M. & Lisiecki, L. E. A Late Pleistocene sea level stack. *Clim. Past* 12, 1079–1092 (2016).

REVIEWERS' COMMENTS:

Reviewer #1 (Remarks to the Author):

The authors have responded adequately to most of the points raised by the reviewers, and revised accordingly. I find that the manuscript can be published in the present form.

Reviewer #2 (Remarks to the Author):

Review of "Model evidence for seasonal bias in Antarctic ice cores" by Erb et al.

Overall the revisions by the authors have improved the paper considerably. I just have a few minor comments that the authors need to address

1) I am very surprised that the largest contributor to the reconstructed temperature changes over Antarctica seems to be the ice-sheet forcing, which, if I understand it correctly, originates mostly from the local changes in ice-sheet height (not from remote effects a la Kawamura 2007). An 8 C temperature change (Figure 2) corresponds approximately to a 1km change in height. This seems to be much higher than the estimates from Jouzel et al. (2007) for EPICA and existing ice-flow models. This needs to be further discussed. A 1 km change in height across Eastern Antarctica would probably lead to an unrealistically large contribution of Antarctica to global sea level variations.

2) The assumption that the temporal variations in Antarctic ice volume are synchronous with the global sea level record needs some further justification and references. The authors refer to the 5 ka lag between obliquity signal in Antarctic and obliquity forcing (line 230), which points to possible delays related to ice-sheet dynamics. If this was the case, the modeling set-up for the Icesheet experiment, which assumes an instantaneous response to global sea level, would be problematic. This needs to be discussed.

3) Lines 182-183: The argument that the direct obliquity forcing and obliquity driven heat transport changes oppose each other has been made in Timmermann et al (2014). This paper should be cited in this context.

Reviewer #3 (Remarks to the Author):

In my opinion, the authors addressed in an appropriate way the concerns I raised in the first review.

REVIEWERS' COMMENTS:

Reviewer #1 (Remarks to the Author):

The authors have responded adequately to most of the points raised by the reviewers, and revised accordingly. I find that the manuscript can be published in the present form.

Reviewer #2 (Remarks to the Author):

Review of "Model evidence for seasonal bias in Antarctic ice cores" by Erb et al.

Overall the revisions by the authors have improved the paper considerably. I just have a few minor comments that the authors need to address

1) I am very surprised that the largest contributor to the reconstructed temperature changes over Antarctica seems to be the ice-sheet forcing, which, if I understand it correctly, originates mostly from the local changes in ice-sheet height (not from remote effects a la Kawamura 2007). An 8 C temperature change (Figure 2) corresponds approximately to a 1km change in height. This seems to be much higher than the estimates from Jouzel et al. (2007) for EPICA and existing ice-flow models. This needs to be further discussed. A 1 km change in height across Eastern Antarctica would probably lead to an unrealistically large contribution of Antarctica to global sea level variations.

The amount of temperature change attributed to ice sheets is determined by idealized simulations run on the GFDL CM2.1 and NCAR CESM models. In these experiments, Last Glacial Maximum ice sheets are prescribed (and sea level is lowered accordingly) with no other changes in forcings. For GFDL CM2.1, ICE-5G (Peltier, 2004) is used. For NCAR CESM, the Paleoclimate Modelling Intercomparison Project Phase III (PMIP3) LGM ice sheets, with a modification to western Labrador Sea ice shelves (Brady, Otto-bliesner, Kay, & Rosenbloom, 2013), are used. These experiments are designed to isolate the influence of ice sheets alone, and both of these experiments have large temperature changes over the ice sheets: averaged from 70-90°S, annual-mean temperature changes due to ice sheets are -8.1°C in GFDL CM2.1 and -7.7°C in NCAR CESM. Here are maps of surface air temperature change due to ice sheets:

Change in surface air temp. ($^{\circ}\text{C}$), IceSheets-preind

Here are maps of the surface height changes:

Change in surface height (m), IceSheets—preind

Additionally, a map of the GFDL CM2.1 temperature anomalies can be seen in Erb et al. 2015, Fig, 1d. Because both GCMs show similar temperature changes in response to LGM-sized ice sheets, it lends confidence to the values presented in the paper. I have added a sentence to the methods to mention which ice sheets are used in these two experiments.

2) The assumption that the temporal variations in Antarctic ice volume are synchronous with the global sea level record needs some further justification and references. The authors refer to the 5 ka lag between obliquity signal in Antarctic and obliquity forcing (line 230), which points to possible delays related to ice-sheet dynamics. If this was the case, the modeling set-up for the Icesheet experiment, which assumes an instantaneous response to global sea level, would be problematic. This needs to be discussed.

Yes, this assumption—that all ice sheet variations are synchronous with global sea level variations—is a simplification which may not be true of the real climate system. For example, evidence suggests that the Antarctic sea ice melt may have lagged changes in other ice sheets (see Fig. 10 of Peltier, 2004), and timing differences like this have the potential to affect the results of this study. I have added a discussion of this limitation to the methods section of the paper. Without better knowledge of the timing of Antarctic ice sheet variations over the past several glacial cycles, however, it is unclear how to address this more fully in the current work. However, the fact that ice sheets may lag the direct obliquity forcing is expected, and not a shortcoming of the experimental design.

3) Lines 182-183: The argument that the direct obliquity forcing and obliquity driven heat transport changes oppose each other has been made in Timmermann et al (2014). This paper should be cited in this context.

I have added this citation.

Reviewer #3 (Remarks to the Author):

In my opinion, the authors addressed in an appropriate way the concerns I raised in the first review.

References

- Brady, E. C., Otto-bliesner, B. L., Kay, J. E., & Rosenbloom, N. (2013). Sensitivity to glacial forcing in the CCSM4. *Journal of Climate*, 26(6), 1901–1925. <https://doi.org/10.1175/JCLI-D-11-00416.1>
- Peltier, W. R. (2004). GLOBAL GLACIAL ISOSTASY AND THE SURFACE OF THE ICE-AGE EARTH: The ICE-5G (VM2) Model and GRACE. *Annual Review of Earth and Planetary Sciences*, 32(1), 111–149. <https://doi.org/10.1146/annurev.earth.32.082503.144359>